# Enhancing the Antimicrobial Properties of Peptides through Cell-Penetrating Peptide Conjugation: A Comprehensive Assessment

**DOI:** 10.3390/ijms242316723

**Published:** 2023-11-24

**Authors:** Sergey V. Kravchenko, Pavel A. Domnin, Sergei Y. Grishin, Nikita A. Vershinin, Elena V. Gurina, Anastasiia A. Zakharova, Viacheslav N. Azev, Leila G. Mustaeva, Elena Y. Gorbunova, Margarita I. Kobyakova, Alexey K. Surin, Roman S. Fadeev, Olga S. Ostroumova, Svetlana A. Ermolaeva, Oxana V. Galzitskaya

**Affiliations:** 1Institute of Environmental and Agricultural Biology (X-BIO), Tyumen State University, 625003 Tyumen, Russia; svkraft@yandex.ru (S.V.K.); syugrishin@gmail.com (S.Y.G.); n.a.vershinin@utmn.ru (N.A.V.); e.v.gurina@utmn.ru (E.V.G.); 2Biology Faculty, Lomonosov Moscow State University, 119991 Moscow, Russia; paveldomnin6@gmail.com; 3Gamaleya Research Centre of Epidemiology and Microbiology, 123098 Moscow, Russia; drermolaeva@mail.ru; 4Institute of Protein Research, Russian Academy of Sciences, 142290 Pushchino, Russia; alan@vega.protres.ru; 5Institute of Cytology, Russian Academy of Sciences, 194064 St. Petersburg, Russia; zaza2187@bk.ru (A.A.Z.); osostroumova@mail.ru (O.S.O.); 6The Branch of the Institute of Bioorganic Chemistry, Russian Academy of Sciences, 142290 Pushchino, Russia; viatcheslav.azev@bibch.ru (V.N.A.); mustaeva@rambler.ru (L.G.M.); eyugorbunova@rambler.ru (E.Y.G.); 7Institute of Theoretical and Experimental Biophysics, Russian Academy of Sciences, 142290 Pushchino, Russia; kobyakovami@gmail.com (M.I.K.); fadeevrs@gmail.com (R.S.F.); 8Research Institute of Clinical and Experimental Lymphology—Branch of the Institute of Cytology and Genetics Siberian Branch of Russian Academy of Sciences, 630060 Novosibirsk, Russia; 9State Research Center for Applied Microbiology and Biotechnology, 142279 Obolensk, Russia

**Keywords:** antimicrobial peptides, Antennapedia peptide, cell-penetrating peptide, TAT fragment, FoldAmyloid, ribosomal S1 protein, amyloid

## Abstract

Combining antimicrobial peptides (AMPs) with cell-penetrating peptides (CPPs) has shown promise in boosting antimicrobial potency, especially against Gram-negative bacteria. We examined the CPP-AMP interaction with distinct bacterial types based on cell wall differences. Our investigation focused on AMPs incorporating penetratin CPP and dihybrid peptides containing both cell-penetrating TAT protein fragments from the human immunodeficiency virus and Antennapedia peptide (Antp). Assessment of the peptides TAT-AMP, AMP-Antp, and TAT-AMP-Antp revealed their potential against Gram-positive strains (*Staphylococcus aureus*, Methicillin-resistant *Staphylococcus aureus* (MRSA), and *Bacillus cereus*). Peptides TAT-AMP and AMP-Antp using an amyloidogenic AMP from S1 ribosomal protein *Thermus thermophilus*, at concentrations ranging from 3 to 12 μM, exhibited enhanced antimicrobial activity against *B. cereus*. TAT-AMP and TAT-AMP-Antp, using an amyloidogenic AMP from the S1 ribosomal protein *Pseudomonas aeruginosa*, at a concentration of 12 µM, demonstrated potent antimicrobial activity against *S. aureus* and MRSA. Notably, the TAT-AMP, at a concentration of 12 µM, effectively inhibited *Escherichia coli* (*E. coli*) growth and displayed antimicrobial effects similar to gentamicin after 15 h of incubation. Peptide characteristics determined antimicrobial activity against diverse strains. The study highlights the intricate relationship between peptide properties and antimicrobial potential. Mechanisms of AMP action are closely tied to bacterial cell wall attributes. Peptides with the TAT fragment exhibited enhanced antimicrobial activity against *S. aureus*, MRSA, and *P. aeruginosa.* Peptides containing only the Antp fragment displayed lower activity. None of the investigated peptides demonstrated cytotoxic or cytostatic effects on either BT-474 cells or human skin fibroblasts. In conclusion, CPP-AMPs offer promise against various bacterial strains, offering insights for targeted antimicrobial development.

## 1. Introduction

The rapid proliferation of antibiotic-resistant bacteria challenges both global health and economic stability [1]. A notable approach to addressing this is the utilization of antimicrobial peptides (AMPs), molecules characterized by extensive antibacterial properties and a reduced propensity to trigger drug resistance [2,3,4,5,6]. Peptides, known for their selectivity and safety, are gaining traction in pharmaceutical research and development, with substantial growth potential and innovative opportunities on the horizon [7,8,9]. Presently, over 30 AMP-associated drugs undergo clinical evaluation, indicating certain AMPs’ capacity to enhance the activity of established antibiotics, even against resistant pathogens [5].

AMPs are emerging as viable alternatives to conventional antibiotics, attributed to their multifaceted antimicrobial effects [10,11,12]. Several studies have reported on the development of synthetic peptide constructs that demonstrate effectiveness against various antibiotic-resistant bacteria, including MRSA, by using a hybrid design [4,13,14,15,16]. One such promising development is the synthetic peptide MPX, which shows notable antibacterial activity, particularly against *E. coli*, and has demonstrated potential in facilitating tissue repair and immune system-driven bacterial clearance in *S. aureus* infections [17]. Such peptides disrupt bacterial growth and compromise cellular membrane integrity, as evidenced by peptide SA6 derived from salt-fermented shrimp [18]. Moreover, they contribute to wound healing, as seen with the synthetic antimicrobial peptide FWKFK [19].

Cell-penetrating peptides (CPPs) are a category of small molecules characterized by robust membrane permeability, facilitating the cellular uptake of peptides, proteins, nucleic acids, and other macromolecules [20]. Antimicrobial peptides (AMPs) containing cell-penetrating peptide (CPP) fragments are of significant relevance in biomedical research due to their unique properties and applications. The inclusion of CPPs enhances cell-penetrating abilities, facilitating efficient intracellular delivery of AMPs and other therapeutic agents. One notable subtype of CPPs is the cationic peptide, marked by the inclusion of basic amino acids like arginine (R) and lysine (K). Arginine-rich CPPs typically consist of 8–25 amino acid residues and are used for drug delivery, biosensors, and vaccines [20]. In our previous research, we tested the TAT fragment (RKKRRQRRR) as a cell-penetrating peptide (CPP) for creating hybrid amyloid-antimicrobial peptides and obtained promising results [4,15]. The antimicrobial effect of hybrid peptides can be shown against both Gram-positive and Gram-negative bacteria [4]. However, it is of interest to investigate other CPPs as components of antimicrobial peptides. It is reasonable to assume that the quality and quantity of the CPP can affect the amyloidogenic properties and antimicrobial effectiveness of hybrid amyloid-antimicrobial peptides. Thus, the aim of the present study was to examine and compare the effects of different CPPs, the positioning of CPPs (at the N- and C-termini of the antimicrobial peptide), and the combined action of two CPPs—a fragment of the TAT peptide and Antp (RQIKIWFQNRRMKWKK). Tat fragment (RKKRRQRRR) and Antennapedia peptide (RQIKIWFQNRRMKWKK) are among the first CPPs that were identified in the 1990s and have since been used as membranotropic molecules, including for the creation of antimicrobial peptides [21,22,23]. Derived from the Antennapedia homeodomain from Drosophila, Antennapedia peptide is a 16-residue peptide [RQIKIWFQNRRMKWKK (43–58)] employed as a vector for the cellular uptake of hydrophilic molecules [21,24,25]. Further results demonstrated that among all tested peptides, Antennapedia peptide is a promising antibacterial peptide with high selectivity and less cytotoxicity for mammalian cells that could be further applied as a delivery system for antibacterial agent delivery [26,27].

Recent studies have shown that conjugation of an antimicrobial peptide with CPP can significantly enhance the peptide’s antimicrobial properties, especially against Gram-negative bacteria [28]. Generally speaking, the distinction between these two types of bacteria, primarily based on their cell wall structure, may influence the effectiveness of CPP-AMPs [29]. Gram-positive bacteria have a substantial peptidoglycan layer, whereas Gram-negative bacteria possess an outer membrane and a thinner peptidoglycan layer [30]. However, to gain a comprehensive understanding of how CPP-AMPs interact uniquely with each type of bacteria, more thorough research would be necessary. Our achievements with the TAT fragment spurred us to design and evaluate antimicrobial peptides incorporating another CPP, Antennapedia peptide. Furthermore, we assessed the antimicrobial properties of dihybrid peptides that integrate two CPPs: specifically, the TAT fragment at the N-terminus and Antennapedia peptide at the C-terminus.

## 2. Results

### 2.1. Design, Construction, Synthesis, and Checking

Previously, we selected amyloidogenic regions in the sequence of S1 ribosomal proteins from various bacteria: VTDFGVFVEI in S1 from *Thermus thermophilus* (*T. thermophilus)* [31] and ITDFGIFIGL in S1 from *P. aeruginosa* [15]. In accordance with the strategy presented in Figure 1, we developed hybrid peptides that contain one or two cell-penetrating peptides (CPPs) in conjunction with an amyloidogenic region, which was predicted using bioinformatics tools such as FoldAmyloid [32], Pasta 2.0 [33], Waltz [34], and AGGRESCAN [35]. FoldAmyloid available online: http://bioinfo.protres.ru/fold-amyloid/ (accessed on 22 August 2023). AGGRESCAN is available online: http://bioinf.uab.es/aggrescan/ (accessed on 23 August 2023). Pasta 2.0 is available online: http://old.protein.bio.unipd.it/pasta2/ (accessed on 23 August 2023). Waltz is available online: http://waltz.switchlab.org/index.cgi (accessed on 23 August 2023). Four glycine residues were used as the linkage sites between the two parts of the peptides.

The TAT fragment (RKKRRQRRR) was used as CPP1, which was attached to the N-terminus of the amyloidogenic peptide, and the Antennapedia peptide fragment (SRQIKIWFQNRRMKWKK) was used as CPP2, which was attached to the C-terminus of the amyloidogenic peptide. To test the possible synergistic effects of using two different penetrating peptides, dihybrid peptides were synthesized that simultaneously contain CPP1 (RKKRRQRRR) at the N-terminus of the amyloidogenic fragment and CPP2 (SRQIKIWFQNRRMKWKK) at the C-terminus of the amyloidogenic peptide (Table 1). CPP1 and CPP2 are attached to the amyloidogenic fragment using linkers of four glycine amino acid residues. Synthesized peptides based on the S1 sequence of *T. thermophilus* (a non-pathogenic microorganism): R23I^T^ (CPP1 + AF), V31K^T^ (AF + CPP2). Synthesized peptides based on the S1 sequence of *P. aeruginosa* (pathogenic microorganism): R23L^P^ (CPP1 + AF), I31K^P^ (AF + CPP2), and R44K^P^ (CPP1 + AF + CPP2). All synthesized peptides were verified for purity (>95%) and sequence matching. Subsequently, the amyloidogenic and antimicrobial properties of the peptides R23I^T^, V31K^T^, R23L^P^, I31K^P^, and R44K^P^ were checked.

### 2.2. Results of the Amyloidogenicity Prediction of Antimicrobial Peptides

In relation to the goal of developing new antimicrobial peptides, which was the synthesis and testing of such amino acid sequence fragments that exert an antimicrobial effect due to directed co-aggregation, we conducted a preliminary prediction of the amyloidogenic properties of such sequences. According to the FoldAmyloid and AGGRESCAN programs, an amyloidogenic fragment IKIWF is predicted in the Antennapedia peptide RQIKIWFQNRRMKWKK (Table 1). Thus, this is an additional second amyloidogenic fragment that is predicted in the peptide sequences V31K^T^, I31K^P^, and R44K^P^.

Only two programs have identified two amyloidogenic sites: FoldAmyloid and AGGRESCAN. Based on these predictions, two amyloidogenic regions were selected in new peptides containing CPP2, one in AF and the other in CPP2.

### 2.3. Thioflavin T Assay

In light of these amyloidogenicity prediction results, we experimentally checked using the amyloid-specific dye thioflavin T (ThT) the ability to form amyloid-like aggregates for the peptides: R23I^T^, V31K^T^, R23L^P^, I31K^P^, and R44K^P^. The corresponding results are presented in Figure 2. ThT, a benzothiazole dye, is known for its unique property of binding to amyloids and amyloid-like fibrils. This interaction results in a substantial increase in the fluorescence intensity of ThT, typically observed at a wavelength of around 485 nm. Fluorescence spectra of ThT were captured after a 24 h incubation period at 37 °C with varying concentrations of the peptides R23I^T^, V31K^T^, R23L^P^, I31K^P^, and R44K^P^.

As can be seen from the data in Figure 2, based on the ThT test for amyloid-like aggregations, there is no significant increase in fluorescence intensity in most peptide samples, even at the maximum peptide concentration of 190–380 μM, depending on the peptide (concentration equivalent to 1 mg/mL). The exception is the peptide R23L^P^, where a multiple increase in the fluorescence intensity of ThT at a wavelength of ~485 nm was detected after incubation for 24 h.

It should be taken into account that the IKIWF fragment might be too small to have a significant effect in favor of fibril formation. At the same time, apparently, increasing the length of the peptide in general and due to the additional CPP leads to a decrease in the ability of peptides to form amyloid-like structures, which was obtained in experiments with ThT and demonstrated in Figure 2. However, high amyloidogenicity is not a determinant; the antimicrobial effect is more essential, which was tested in subsequent experiments testing the synthesized peptides against various pathogenic bacteria.

### 2.4. Antibacterial Activity of Peptides R23I^T^ and V31K^T^ against Gram-Negative and Gram-Positive Bacteria in Liquid Medium

We evaluated the antimicrobial effects of R23I^T^ and V31K^T^ peptides in liquid media against Gram-negative bacteria, specifically *P. aeruginosa* (ATCC 28753 strain) and *E. coli* (K12 strain). In addition, we assessed the antimicrobial activity of the R23I^T^ and V31K^T^ peptides against Gram-positive bacteria, including *S. aureus* (209P strain), MRSA (ATCC 43300 strain), and *B. cereus* (IP 5832 strain). The results of these tests are presented in Figure 3 and Appendix A.

As shown in Figure 3, R23I^T^ and V31K^T^ peptides showed no significant antimicrobial effects (comparable to gentamicin) after 15 h of incubation against *P. aeruginosa* (ATCC 28753 strain) and *E. coli* (K12 strain). In contrast, the R23I^T^ peptide at a concentration of 12 μM showed the effect of inhibiting the growth of *S. aureus* (209P strain). At the same time, there were no significant inhibitory effects against MRSA (comparable to gentamicin) after 15 h for the R23I^T^ and V31K^T^ peptides. Both peptides showed an inhibitory effect against *B. cereus*. Thus, peptides R23I^T^ and V31K^T^ at concentrations of 3–12 μM significantly inhibited growth, demonstrating antimicrobial effects comparable to those of gentamicin.

### 2.5. Antibacterial Activity of Peptides R23L^P^, V31K^P^, and R44K^P^ against Gram-Positive and Gram-Negative Bacteria in Liquid Medium

We further tested the R23L^P^, I31K^P^, and R44KP peptides, which were designed to contain an amyloidogenic fragment from the *P. aeruginosa* S1 protein, as well as one or two cell penetrating peptide (CPP) fragments. We checked the antimicrobial effects of peptides R23L^P^, I31K^P^, and R44K^P^ in a liquid medium against Gram-negative bacteria, including *P. aeruginosa* (ATCC 28753 strain) and *E. coli* (K12 strain). In addition, we evaluated the antimicrobial effects of R23L^P^, I31K^P^, and R44K^P^ peptides against Gram-positive *S. aureus* (209P strain), MRSA (ATCC 43300 strain), and *B. cereus* (IP 5832 strain) (Figure 4 and Appendix A).

As shown in Figure 4, the range of concentrations at which the peptides showed an antimicrobial effect against *P. aeruginosa* (ATCC 28753 strain) was 6–12 µM for the R23L^P^ peptide and 6–12 µM for the R44K^P^ peptide. Antimicrobial effects comparable to gentamicin were not found for the I31K^P^ peptide against *P. aeruginosa* (ATCC 28753 strain). Growth inhibition of *E. coli* (K12 strain) cells during 15 h of incubation, similar to the antimicrobial effect of the positive control antibiotic gentamicin, was observed only for samples with the R23L^P^ peptide at a concentration of 12 µM. At the same time, the R23L^P^ and R44K^P^ peptides at a concentration of 12 µM had an antimicrobial effect on *S. aureus* (209P strain) and MRSA (ATCC 43300 strain) cells comparable to gentamicin, in contrast to the I31K^P^ peptide, which did not show strong effects against these microorganisms. The strongest antimicrobial effect of the peptides was observed against *B. cereus* (IP 5832 strain) after incubation for 15 h in the concentration range: 1.5–12 µM for the R23L^P^ peptide, 12 µM for the I31K^P^ peptide, and 3–12 µM for the R44K^P^ peptide.

### 2.6. Peptides’ Antibacterial Activity against S. aureus, MRSA, P. aeruginosa, and E. coli

We tested the antimicrobial activity of the peptides against various strains of *S. aureus*, MRSA, *P. aeruginosa*, and *E. coli* bacteria. In addition to the data described above obtained for susceptible strains 209P (*S. aureus*), ATCC 43300 (MRSA), ATCC 28753 (*P. aeruginosa*), and K12 (*E. coli*), results of peptide testing for their antimicrobial effect against clinical isolate strains 129B (*S. aureus*), SA 180-F (MRSA), and 2943 (*P. aeruginosa*), and susceptible MG1655 (*E. coli*) were also obtained (Appendix A). For comparison, Table 2 shows data for two strains of each organism.

According to Table 2, and as a result of comparing the effects of peptides on two different strains of each of the microorganisms (*S. aureus*, MRSA, *E. coli*, and *P. aeruginosa*), it can be concluded that the highest antimicrobial effect was shown by peptides R23L^P^ and R44K^P^.

### 2.7. Effect of R23I^T^ Peptide on Membrane Ionic Permeability

The most common anionic lipid in bacteria is phosphatidylglycerol, while phosphatidylethanolamine is the most abundant among the zwitterionic lipids in bacteria [36,37,38]. Thus, to mimic the membranes of target microorganisms, an equimolar mixture of POPE and POPG was used. Figure 5 shows examples of typical records of the time course of the transmembrane current (*I*) flowing through the bilayers composed of POPE/POPG (50:50 mol%) in the absence of the R23I^T^ peptide and in response to a subsequent increase in the concentration of peptide in the bathing solution. Figure 5A demonstrates that an intact bilayer was not permeable to ions. The one-side addition of R23I^T^ peptide up to a concentration of 80 μg/mL (29.9 µM) did not cause an increase in the conductance of the POPE/POPG (50:50 mol%) membrane (Figure 5B).

At the same time, a subsequent increase in the concentration of the R23I^T^ peptide (>80 µg/mL or the same >29.9 µM) led to a disturbance of the electrical stability of the membrane and its subsequent disruption. The results obtained indicate that the R23I^T^ peptide might exhibit a detergent effect on the lipid bilayers that mimic the bacterial membranes. As a positive control, similar experiments were carried out with a well-known detergent, Triton X-100, and pore-forming antimicrobial peptides, magainin I, gramicidin A, and duramycin [39,40,41,42,43,44] (Appendix A).

### 2.8. Toxicity of V31K^T^, I31K^P^, and R44K^P^ against Eukaryotic Cells

Figure 6 shows the results of a study of the viability of human skin fibroblasts after incubation with peptides for 96 h. It was shown that all the studied peptides did not reduce cell viability.

To analyze the cytotoxic effect, cells after incubation with peptides were stained with calcein AM (stains living cells) and propidium iodide (stains dead cells). It was shown that the studied peptides, at a concentration of 50 µg/mL (both with and without pre-incubation), had no cytotoxic effect on BT474 cells or human skin fibroblasts after 96 h of incubation.

Analysis of the cytostatic effect was performed on the basis of the distribution of cells by phases of the cell cycle and mitotic activity. It was shown that all the studied peptides, at a concentration of 50 μg/mL (both with and without pre-incubation), did not suppress the proliferative activity of BT-474 cells and human skin fibroblasts. Additionally, using the HemoPred Internet resource (http://codes.bio/hemopred/ (accessed on 2 November 2023)), it was predicted that the peptides R23I^T^, V31K^T^, and R44K^P^ do not have hemolytic activity, and the peptides R23L^P^ and I31K^P^ may have hemolytic activity. At the same time, R23L^P^ and I31K^P^ peptides not only lacked cytotoxic effects on human nucleated cells such as fibroblasts and epithelial cells of breast cancer, but also did not even suppress their proliferative activity. Therefore, we assume that the R23L^P^ and I31K^P^ peptides may have little hemolytic activity. However, these features of the peptides require further study. The cytotoxic effect of the peptides R23I^T^ and R23L^P^, as well as their analogs, was tested in previous similar studies [15,31]. They have not been shown to have significant effects over the tested concentration range.

## 3. Discussion

The underlying mechanisms of antimicrobial resistance stand as critical concerns necessitating immediate intervention [45]. Key mechanisms include the production of enzymes that degrade or modify the drug, alteration of the drug target site, changes in membrane permeability that restrict drug access, and the activation of efflux pumps that expel the drug from the microbial cell [46,47,48]. Over time, these adaptations can render treatments ineffective, leading to persistent infections and an increased potential for disease transmission. Integrating cell-penetrating peptide (CPP) fragments into AMPs has emerged as a significant area of interest in biomedical research due to their unique properties and applications [20,49]. Notably, AMPs pose a lower risk of resistance development in microorganisms, as they target fundamental bacterial components. The flexibility to design and engineer AMPs for improved stability, specificity, and activity, coupled with CPPs for enhanced penetration and targeted delivery, underlines their versatility. Thus, AMPs with CPP fragments represent a burgeoning research field with substantial potential for developing novel therapeutic and diagnostic strategies. By incorporating CPPs, the cell-penetrating abilities of AMPs are enhanced, enabling effective intracellular delivery of AMPs and other therapeutic agents. The ability to design and engineer AMPs for improved stability, specificity, and activity, combined with the utilization of CPPs for superior penetration and targeted delivery, underscores their versatility.

Recently, the use of specialized software to predict amyloidogenic regions in proteins and peptides has become a common practice before experimentally testing their amyloidogenic properties [50,51]. Certain regions or domains in proteins and peptides, known as ‘hotspots’, are primarily responsible for initiating the formation of amyloid fibrils. In this context, it can be assumed that the IKIWF fragment, predicted by the FoldAmyloid [32] and Waltz [34] programs due to its limited length, contributes minimally to the amyloidogenesis of peptides. The overall length, conformation, and surrounding sequence of the peptide likely play a pivotal role in modulating these amyloidogenic tendencies. While the R23L^T^ peptide indeed exhibited a significant fluorescence increase, suggesting the formation of an amyloid structure, the general trend among other peptides of varying lengths did not present a consistent pattern that would confidently establish a direct correlation between length and amyloidogenicity. It is also pertinent to note that the elongation of peptides, especially with the incorporation of an additional CPP, seemingly reduces their propensity to adopt amyloid structures. If this trend is corroborated by more extensive datasets, it might imply that the addition of specific sequences or domains could introduce structural constraints, thereby tempering amyloidogenic tendencies.

An intriguing facet of our study revolves around the strain specificity exhibited by the peptides R23I^T^ and V31K^T^. When assessing the activity of antimicrobial agents, it is imperative to consider not only their broad-spectrum activity but also their specificity towards individual bacterial strains. In our experiments, the peptide R23I^T^ demonstrated an appreciable inhibitory effect on the growth of the *S. aureus* (209P strain) at a concentration of 12 μM. Contrastingly, both R23I^T^ and V31K^T^ displayed a lack of significant activity against the MRSA strain. This suggests that the peptides might possess an intrinsic specificity towards certain bacterial strains, rendering them more effective against some pathogens while being less potent against others. The observed strain specificity raises several pertinent questions: Do these peptides target specific molecular components unique to certain bacterial strains? Might there be differences in membrane composition or surface proteins between the strains that influence peptide activity? Further investigations are warranted to delve deeper into these aspects. However, the current findings underscore the importance of considering strain-specific responses when evaluating the therapeutic potential of antimicrobial peptides.

In the course of our investigation, we evaluated the antimicrobial effects of multiple peptides, namely, R23I^T^, V31K^T^, R23L^P^, I31K^P^, and R44K^P^, against various Gram-positive bacterial strains, including *S. aureus*, MRSA, and *B. cereus*. Notably, the R23I^T^ peptide exhibited significant inhibitory activity against *S. aureus* (209P strain) at a concentration of 12 μM, while both R23I^T^ and V31K^T^ peptides demonstrated limited inhibitory effects against MRSA and strong inhibition against B. cereus. This corroborates their potential as antimicrobial agents. Additionally, R23LP and R44KP peptides displayed remarkable antimicrobial activity comparable to gentamicin against *S. aureus* (209P strain) and MRSA (ATCC 43300 strain), with R23L^P^ being particularly effective against *B. cereus*. Further assessment against Gram-negative bacteria (*E. coli* and *P. aeruginosa*) revealed varying outcomes. Interestingly, the R23L^P^ peptide demonstrated significant growth inhibition against *E. coli* (K12 strain) at a 12 μM concentration. Tests of strains with different media in different institutes were performed. The influence of the media remains to be checked in the future. It should be mentioned that clinically achievable concentrations are unknown, and threshold concentrations of activity against several bacteria used remain unknown and should be tested in the future.

Overall, our findings suggest that the length and composition of the peptides play a crucial role in determining their antimicrobial potential against different bacterial strains. Further exploration is warranted to fully unravel the mechanisms underlying these strain-specific effects and to harness them for targeted antimicrobial interventions.

In the context of cell cycle phase distribution and mitotic activity analysis, it was observed that, when administered at a concentration of 50 μg/mL, all the examined peptides (with and without pre-incubation) did not exert any inhibitory effects on the proliferative activity of both BT-474 cells and human skin fibroblasts. In summary, the results obtained in this study affirm that the tested peptides did not compromise the viability of BT-474 cells and human skin fibroblasts. Furthermore, they did not induce any cytotoxic or cytostatic effects on these cell lines. We believe that the main reason for the lack of toxicity of the studied AMP in relation to mammalian cells is due to the different structure of the cell wall in prokaryotic and eukaryotic cells. In particular, teichoic acids in the cell wall of Gram-positive bacteria and LPS in the outer membrane of Gram-negative bacteria impart an electronegative charge to the surface of microorganisms, enhancing interaction with AMP. At the same time, the membrane layer of eukaryotic cells contains phosphatidylcholine and sphingomyelin, which prevent interaction with AMP due to their neutral charge at physiological pH [52]. Interestingly, the potential hemolytic activity of the peptides R23L^P^ and I31K^P^ studied in the work was predicted using in silico methods. At the same time, in an in vitro study, these peptides were not toxic against human fibroblasts and epithelial cells closest to human erythrocytes. We assume that this may indicate the potentially low hemolytic activity of these peptides. In silico methods for predicting hemolytic activity are certainly of great importance for the primary screening of AMP. However, the accuracy of predicting the potential hemolytic activity of peptides can vary greatly depending on the features of their structure. Thus, in the work of Robles-Loaiza et al., all available in silico models for predicting the hemolytic activity of peptides are described most fully [53]. The authors convincingly prove the need to develop in silico methods for predicting toxicity and effective development of peptides. However, as stated in this paper, the accuracy of the studied models for predicting the toxicity of peptides is still about 50%. Therefore, in silico methods must necessarily be supplemented with in vitro and, preferably, in vivo studies.

Generally, the mechanisms by which AMPs act against bacteria depend on the properties of the bacterial cell wall. Gram-positive bacteria have a thick peptidoglycan layer, while Gram-negative bacteria have an outer membrane and a thinner peptidoglycan layer, which can impact the effectiveness of different antimicrobial strategies [54,55]. Interestingly, only peptides containing the TAT fragment as CPP showed the greatest antimicrobial effect against *S. aureus*, MRSA, and *P. aeruginosa*. If the peptide contained only the Antp fragment, then, in general, its antimicrobial activity was lower compared to those peptides that contained the TAT fragment as CPP (in our case, CPP1 + AF and CPP1 + AF + CPP2). Only amyloidogenic fragments (VTDFGVFVEI from S1 *T. thermophilus* and ITDFGIFIGL from S1 *P. aeruginosa,*
Table 2) demonstrated antibacterial activity at a higher concentration of 913 µM ([31], Table 2, and Appendix A).

Peptides can exert their effects not only through membrane disruption but also via mechanisms such as bactericidal protein aggregation, protein function inhibition, and pathogen agglutination. Primarily, this pertains to amyloidogenic peptides. Amyloidogenic peptides, which have often been associated with neurodegeneration, are now recognized as important players in combating microbial infections [56]. Innovative strategies for antimicrobial action have also been explored, such as “protein silencing” based on directed co-aggregation, where the amyloidogenic antimicrobial peptide interacts with the target protein of bacteria and forms aggregates, effectively disabling the protein [57]. Additionally, moonlighting ribosomal proteins, capable of multiple biochemical functions, including inhibiting various pathogens, could offer a solution to the issue of antibiotic resistance [58]. The field of AMPs, both natural and synthetic, holds immense potential in combating pathogens and overcoming antibiotic resistance. Through molecular design and genetic engineering approaches, new AMP drugs are being developed, which could revolutionize the treatment of microbial infections and drug-resistant bacteria [11,59]. Researchers hope that these peptides will open new avenues in the fight against antibiotic resistance. However, the development and application of AMPs are not without challenges. The extraction difficulties, high costs, stability issues, and biosafety concerns are some of the major roadblocks to their widespread use [60]. To address these, various strategies are being explored, such as computational prediction, molecular dynamics simulation, gene editing technologies, and improved industrial infrastructure [55,60,61]. Moreover, AMPs are now being recognized for their roles beyond being antimicrobial agents. They have been found to regulate host immune responses, wound healing, and apoptosis, broadening their potential therapeutic applications [61].

## 4. Materials and Methods 

### 4.1. Design and Synthesis of Hybrid Peptides with Cell-Penetrating Peptides and Amyloidogenic Regions

Peptides RKKRRQRRRGGGGVTDFGVFVEI (R23I^T^, molecular weight = 2675.1 Da), VTDFGVFVEIGGGGSRQIKIWFQNRRMKWKK (V31K^T^, molecular weight = 3669.3 Da), RKKRRQRRRGGGGITDFGIFIGL (R23L^P^, molecular weight = 2645.1 Da), ITDFGIFIGLGGGGSRQIKIWFQNRRMKWKK (I31K^P^, molecular weight = 3639.3 Da), and RKKRRQRRRGGGGITDFGIFIGLGGGGSRQIKIWFQNRRMKWKK (R44K^P^, molecular weight = 5189.1 Da) were commercial products (IQ Chemical LLC, S. Petersburg, Russia). The appropriate fractions were lyophilized, and the peptide identity was confirmed using an Orbitrap Elite mass spectrometer (Thermo Scientific, Dreieich, Germany). The estimated peptide molecular weight coincided with the calculated one.

### 4.2. Prediction of Amyloidogenicity of Antimicrobial Peptides

Amyloidogenic regions of protein S1 from *T. thermophilus* and protein S1 from *P. aeruginosa* with a length of at least five amino acid residues were determined using four programs—AGGRESCAN, FoldAmyloid, Pasta 2.0, and Waltz [31,62]. Predicted amyloidogenic regions were checked in vitro by their ability to form amyloid-like aggregates, and sequences VTDFGVFVEI and ITDFGIFIGL were used as amyloidogenic fragments for the R23I^T^, V31K^T^ and R23L^P^, I31K^P^, R44K^P^ peptides, respectively.

### 4.3. Thioflavin T Fluorescence Measurement

Preparations of the R23I^T^, V31K^T^, R23L^P^, I31K^P^, and R44K^P^ peptides in buffer conditions 50 mM TrisHCl, pH 7.5; 150 mM NaCl, 20% (*v*/*v*) DMSO was incubated with 200 µM thioflavin T (ThT) at 37 °C with shaking for 24 h at 450 rpm in a thermostatic mixer, Thermomixer comfort (Eppendorf, Hamburg, Germany). The spectra of fluorescence intensity of free ThT and in solution with individual peptides were studied by us using the method of fluorescence spectroscopy as described previously [62].

### 4.4. Strains of Microorganisms

Two *S. aureus* strains were used: a susceptible strain 209P and a clinical isolate strain 129B (resistant to antibiotics: benzylpenicillin, clindamycin, erythromycin, oxacillin, sulfamethoxazole, and vancomycin). Two MRSA strains were included: a susceptible strain ATCC 43300 (resistant to antibiotics: methicillin, oxacillin (beta-lactam antibiotics), ampicillin (https://www.atcc.org/products/43300 (accessed on 11 November 2023)) and a clinical isolate strain SA 180 (resistant to antibiotics: benzylpenicillin and oxacillin (beta-lactam antibiotics), ciprofloxacin, clindamycin, erythromycin, levomycetin, sulfamethoxazole, and vancomycin). One *B. cereus* was used: a susceptible strain, IP 5832. Two *P. aeruginosa* strains were included: a susceptible strain, ATCC 28753, and a resistant strain, 2943. Two *E. coli* strains were used: a susceptible strain, K12, and a susceptible strain, MG1655.

### 4.5. Measurement of the Antibacterial Activity of Peptides against S. aureus, MRSA, and P. aeruginosa in Liquid Medium

The R23I^T^, V31K^T^, R23L^P^, I31K^P^, and R44K^P^ peptides and gentamicin sulfate were dissolved in 100% DMSO, and solutions were used with a final DMSO concentration of 2% (*v*/*v*) to measure their activity. Determination of the growth for the bacteria *S. aureus* strain 209P, MRSA strain ATCC 43300, *B. cereus* strain IP 5832, *P. aeruginosa* strain ATCC 28753, and *E. coli* strain K12 was performed using Mueller–Hinton Broth (MHB) (Sigma-Aldrich, St. Louis, MO, USA). Bacteria of *S. aureus* (strain 129B) and MRSA (strain SA 180-F) were grown on the brain-heart infusion (BHI) medium, and bacteria of *P. aeruginosa* (strain 2943) and *E. coli* (strain MG1655) were grown on the LB medium. The preparation and co-incubation of peptides with bacteria were carried out as described previously [15]. The samples were grown at 37 °C.

### 4.6. Electrophysiological Assay

An electrophysiological method was used to investigate the possible action of the R23I^T^ peptide on the planar lipid bilayers that mimic bacterial membranes. The model membranes were formed according to the Montal and Mueller technique [63]. Taking into account the efficiency of R23I^T^ peptide against both Gram-negative and Gram-positive bacteria, an equimolar mixture of 1-palmitoyl-2-oleoyl-sn-glycero-3-phosphoethanolamine (POPE) and 1-palmitoyl-2-oleoyl-sn-glycero-3-phospho-(1′-rac-glycerol) (POPG) was chosen to form the lipid bilayers. The lipids were obtained from Avanti Polar Lipids^®^ (Avanti Polar Lipids, Inc., Alabaster, AL, USA). Two symmetrical compartments of the Teflon chamber (*out* and *in*) were divided by 10 µm thick Teflon film with an aperture of 50 µm in diameter. Hexadecane was used for the aperture pretreatment. Membranes were bathed in 0.1 M KCl and 10 mM HEPES at pH 7.4. The R23I^T^ peptide was added to the bathing solution of the *out*-side compartment up to a concentration in the range of 5 to 80 μg/mL from DMSO stock solutions after bilayer formation. As a positive control, we used the detergent Triton X-100, as well as the pore-forming antimicrobial peptides magainin I, gramicidin A, and duramycin. These membrane-active agents were obtained from Sigma-Aldrich Company Ltd. (Sigma-Aldrich, Gillingham, UK).

To apply the transmembrane voltage (*V*) and measure the current flowing through the bilayer (*I*), Ag/AgCl electrodes with 1.5% agarose/2 M KCl bridges were used. “Positive voltage” refers to the case where the *out*-side compartment is positive with respect to the *in*-side. The transmembrane current was measured using an Axopatch 200B amplifier (Molecular Devices, LLC, Orleans Drive, Sunnyvale, CA, USA) under voltage clamp conditions. Data were digitized by a Digidata 1550A and analyzed by pClamp 10.7 (Molecular Devices, LLC, Orleans Drive, Sunnyvale, CA, USA). Data were acquired at a sampling frequency of 5 kHz using low-pass filtering at 1 kHz, and the current tracks were processed using an 8-pole Bessel 100-kHz filter.

### 4.7. Determination of Toxicity of Peptides against Eukaryotic Cells

BT474 human breast duct carcinoma cells were obtained from the American Collection of Type Cell Cultures (Manassas, Virginia, VA, USA). BT474 cells were cultured in DMEM medium (Sigma-Aldrich, St. Louis, MO, USA) supplemented with 10% fetal calf serum (Gibco, Waltham, MA, USA) and 40 μg/mL gentamicin sulfate (Sigma-Aldrich, St. Louis, MO, USA) at 37 °C under conditions of 5% CO_2_ in the air.

The culture of normal human fibroblasts was obtained from the skin areas of the inner surface of the forearm of healthy volunteers after their consent using the explant method. All volunteers gave their informed consent for inclusion before they participated in the study. The study was conducted in accordance with the Declaration of Helsinki, and the protocol was approved by the Ethics Committee of ITEB RAS No. 43/2023. The culture of normal human fibroblasts was obtained from the skin areas of the inner surface of the forearm using the explant method. A 5 × 5 mm skin explant was washed for several minutes in phosphate buffer containing 400 μg/mL gentamicin sulfate. The tissue was mechanically crushed into fragments 1–2 mm^2^ in size, then the fragments were attached to the surface of the culture dish, dried in air for 5–7 min, then covered with alfa MEM medium with 10% fetal calf serum and incubated at 37 °C under conditions of 5% CO_2_ content in the air. The medium around the fragments was changed every 3–4 days. After the release of cells from the fragments, the cells were detached from the surface of the culture plastic, and the number of cells was determined. Cells from passages 1–4 were used in the experiments. Cell subculturing was carried out in alfa MEM medium with 10% fetal calf serum and 40 μg/mL gentamicin sulfate at 37 °C under conditions of 5% CO_2_ in the air.

Cell viability after incubation with cytotoxic agents was assessed by the intensity of resazurin reduction (Sigma-Aldrich, St. Louis, MO, USA). For this, resazurin at a concentration of 30 μg/mL was added to the cells after incubation with substances. At least 5 × 10^4^ cells were examined for each concentration of peptides. Next, the cells were incubated with the dye for 4 h at 37 °C under conditions of 5% CO_2_ in air, and the fluorescence intensity was measured at an excitation wavelength of 532 nm and an emission wavelength of 590 nm using an Infinite F200 plate reader (Tecan, Grödig, Austria). All measurements were performed against an untreated control (cells without the addition of peptides).

The number of living and dead cells was determined by staining cells with the fluorescent dyes calcein AM (Sigma-Aldrich, St. Louis, MO, USA) and propidium iodide (Sigma-Aldrich, St. Louis, MO, USA). Cells were detached from the surface of the samples using an Accutase cocktail. Cells were stained in L-15 medium with 1% fetal bovine serum containing 1 μg/mL calcein AM and 2 μg/mL propidium iodide for 25 min at 37 °C. Live and dead cells were analyzed using a BD Accuri C6 flow cytometer (BD Bioscience, San Jose, CA, USA).

To study the distribution of cells over the phases of the cell cycle, cells were collected, washed in phosphate-buffered saline (300 g, 5 min), and fixed with 70% ethanol (24 h, −20 °C). Staining was carried out for 15 min at 37 °C in the following medium: phosphate-buffered saline, 0.1% Triton X100, 10 μg/mL propidium iodide, and 100 μg/mL RNase (Biolot, St. Petersburg, Russia). The analysis was performed using a BD Accuri C6 flow cytometer (BD Bioscience, San Jose, CA, USA). The obtained results were processed in the ModFit LT 4.1 program.

To determine the mitotic activity, the cells were washed in phosphate-buffered saline (300 g, 5 min) and fixed with 70% ethanol (24 h, −20 °C). Then, they were stained with bisBenzimide H 33342 fluorescent dye at a concentration of 1 μg/mL for 10 min at room temperature, and the number of mitotic cells was counted using a DM 6000 fluorescent microscope (Leica, Wetzlar, Germany). At least 500 cells were analyzed.

### 4.8. Statistical Analysis

The SigmaPlot 14.5 software package (SPSS 14.5, SPSS Inc., Chicago, IL, USA) was used for statistical analysis. The results were expressed as mean ± standard deviation (M ± SD). The experiments were performed in at least two repetitions (*n* ≥ 2). The statistical significance of the difference was determined using analysis of variance (ANOVA) and the Student’s *t*-test.

## 5. Conclusions

In conclusion, AMPs integrated with CPP fragments represent a burgeoning research field with significant potential for the development of innovative therapeutic and diagnostic strategies. Certain peptides (R23I^T^, V31K^T^, R23L^P^, I31K^P^, and R44K^P^) show promise against Gram-positive bacteria (*S. aureus*, MRSA, and *B. cereus*). Peptides R23I^T^ and V31K^T^ are effective against *B. cereus* at concentrations of 3 to 12 μM. R23L^P^ and R44K^P^ are potent against *S. aureus* and MRSA at 12 μM. R23L^P^, at 12 μM, effectively inhibits *E. coli* growth. Peptides with the TAT fragment as CPP perform better against certain bacteria. To sum up, although a broad-spectrum antimicrobial effect is frequently considered favorable, the strain-specific effects of the R23I^T^, V31K^T^, R23L^P^, I31K^P^, and R44K^P^ peptides can be utilized for precise interventions, particularly in the context of addressing infections caused by distinct bacterial strains. At the same time, the in silico prediction of the hemolytic activity of some AMP used in the work remains an open question, since the same AMP did not have toxicity against human nucleated cells. However, this issue requires further careful study in terms of preventing potential systemic toxicity. Future studies should focus on elucidating the molecular mechanisms underpinning this specificity, which would pave the way for designing tailored peptide-based therapeutics.

## Figures and Tables

**Figure 1 ijms-24-16723-f001:**
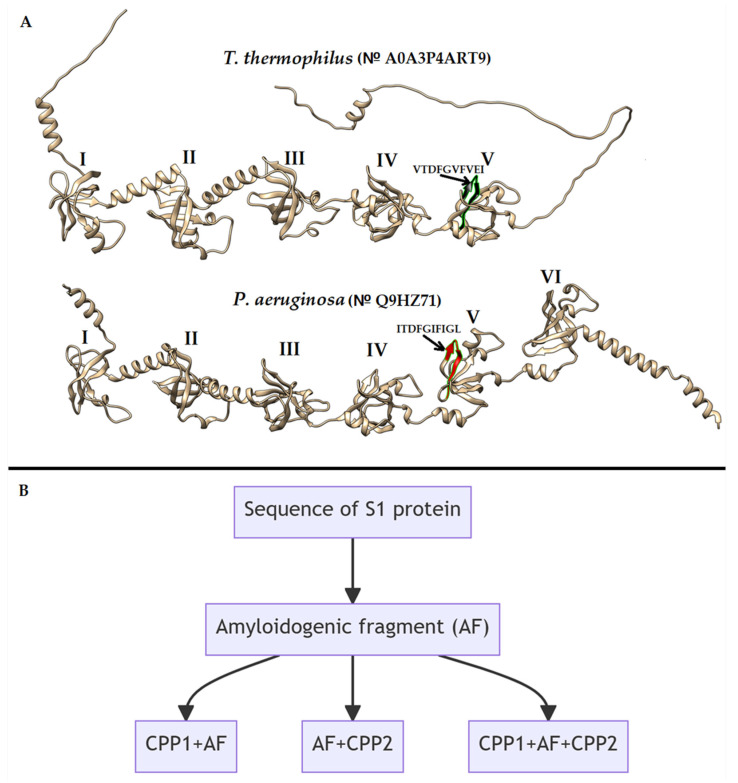
Design scheme for antimicrobial peptides based on amyloidogenic fragments (AFs) of S1 ribosomal proteins and cell penetrating peptides (CPPs). S1 ribosomal protein structures obtained from the AlphaFold Protein Structure Database (https://alphafold.ebi.ac.uk/ (accessed on 28 June 2023)), arrows indicate a special fragment of sequence containing AF; I–VI denote S1 domains in the structure of S1 proteins (**A**). Possible configurations of AF with cell penetrating peptide containing fragment TAT (CPP1) and Antennapedia peptide fragment (CPP2) (**B**).

**Figure 2 ijms-24-16723-f002:**
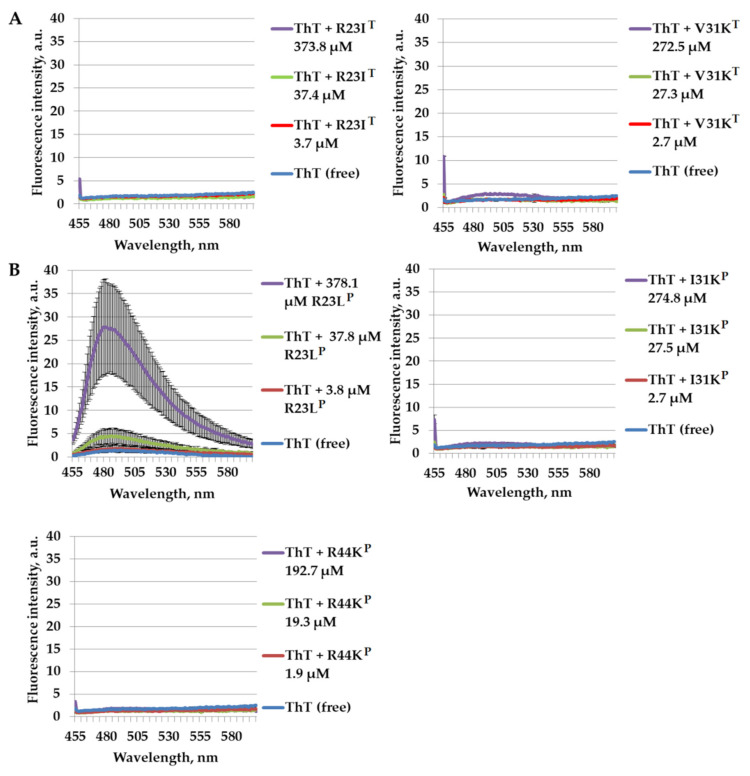
The results of measuring the intensity of thioflavin T (ThT) fluorescence in peptide preparations of R23I^T^ and V31K^T^ based on amyloidogenic fragment (AF) of VTDFGVFVEI of S1 from *T. thermophilus* (**A**), R23L^P^, V31K^P^, and R44K^P^ based on AF of ITDFGIFIGL of S1 from *P. aeruginosa* (**B**) after 24 h of incubation are shown. Error bars show standard errors.

**Figure 3 ijms-24-16723-f003:**
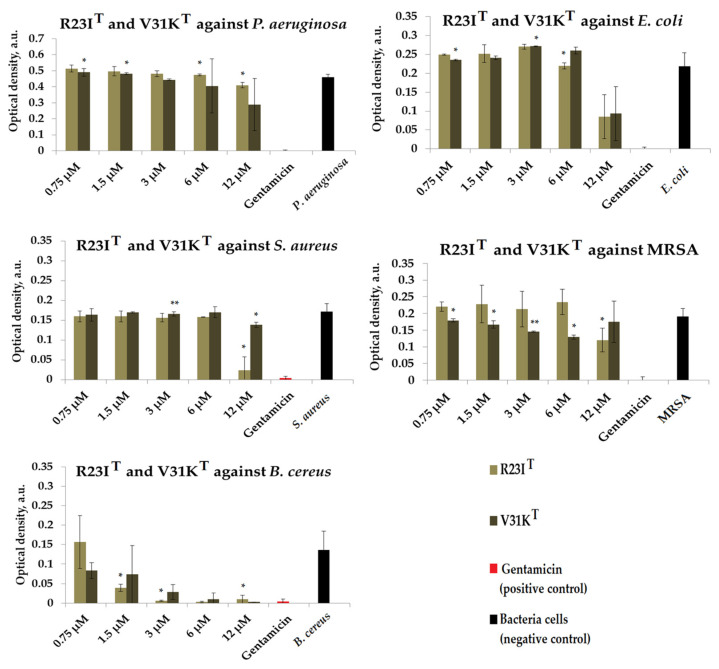
Results of R23I^T^ and V31K^T^ tests for antimicrobial effect against *P. aeruginosa* (ATCC 28753 strain), *E. coli* (K12 strain), *S. aureus* (209P strain), MRSA (ATCC 43300 strain), and *B. cereus* (IP 5832 strain) after a 15 h incubation period. The antibiotic gentamicin was used as a positive control. The concentration of gentamicin was 200 µM. Cell cultures in a liquid medium (Mueller–Hinton) served as a negative control. Error bars show the standard deviation, number of independent experiments are two. *—for significant differences with control *p* < 0.05. **—for significant differences with control *p* < 0.01.

**Figure 4 ijms-24-16723-f004:**
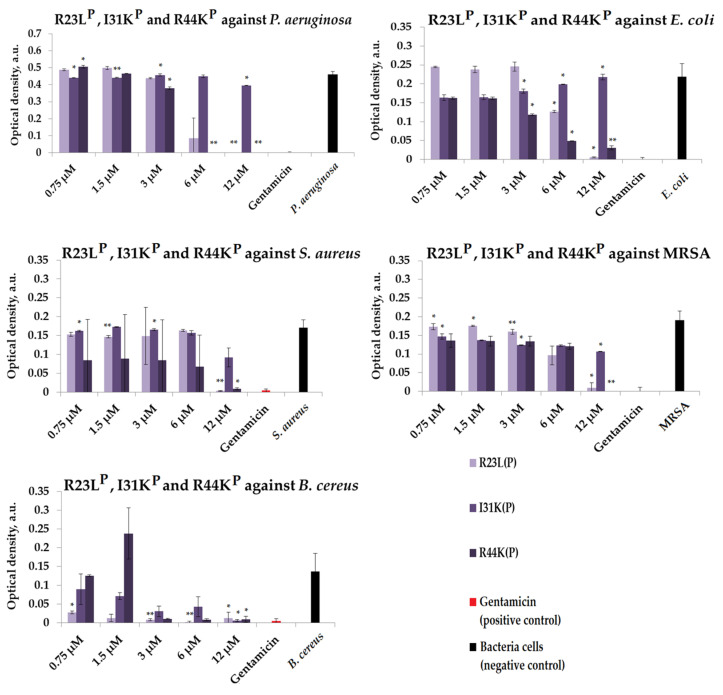
Results of R23L^P^, I31K^P^, and R44K^P^ tests for antimicrobial effect against *P. aeruginosa* (ATCC 28753 strain), *E. coli* (K12 strain), *S. aureus* (209P strain), MRSA (ATCC 43300 strain), and *B. cereus* (IP 5832 strain) after a 15 h incubation period. The antibiotic gentamicin was used as a positive control. The concentration of gentamicin was 200 µM. Cell cultures in a liquid medium (Mueller–Hinton) served as a negative control. Error bars show the standard deviation, number of independent are two. *—for significant differences with control *p* < 0.05. **—for significant differences with control *p* < 0.01.

**Figure 5 ijms-24-16723-f005:**
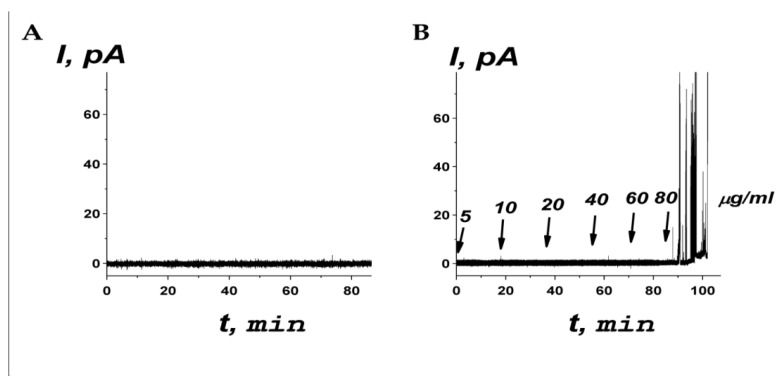
Typical time courses of the transmembrane current flowing through the bilayer in the absence (**A**) and after addition of the R23I^T^ peptide up to the different concentrations in the bathing solution (**B**). The concentrations 5, 10, 20, 40, 60, and 80 µg/mL correspond to 1.9, 3.7, 7.5, 15.0, 22.4, and 29.9 µM. The bilayers were formed from an equimolar mixture of POPE and POPG (50:50 mol%) and bathed in 0.1 M KCl, pH 7.4. The transmembrane voltage was equal to 100 mV. The moments of R23I^T^ peptide addition are indicated by arrows. The corresponding concentrations of the peptide in the membrane-bathing solution are shown above the arrows.

**Figure 6 ijms-24-16723-f006:**
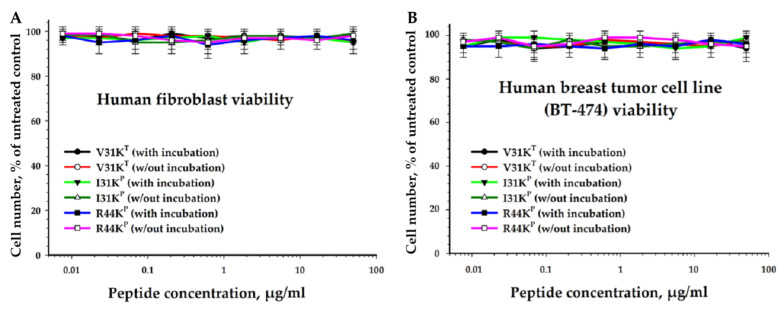
Effects of peptide treatment on survival of human fibroblasts (**A**) and breast tumor cell line BT-474 (**B**). Error bars show that the standard deviation and number of independent experiments are two.

**Table 1 ijms-24-16723-t001:** Predicted amyloidogenic properties of peptides.

Peptide	FoldAmyloid	Waltz	AGGRESCAN	Pasta 2.0
Peptides based on the S1 sequence of *T. thermophilus*				
RKKRRQRRRGGGGVTDFGVFVEI ^§^ (CPP1 + AF, R23I^T^), 23 a.a.	GVFVEI (18–23 a.a.)	No	TDFGVFVEI (15–23 a.a.)	VTDFGVFV (14–21 a.a.)
VTDFGVFVEIGGGGSRQIKIWFQNRRMKWKK (AF + CPP2, V31K^T^), 31 a.a.	GVFVE (5–9 a.a.), IKIWFQ (18–23 a.a.)	VFVEIG (6–11 a.a.)	DFGVFVEI (3–10 a.a.), IKIWF (18–22 a.a.)	VTDFGVFV (1–8 a.a.)
Peptides based on the S1 sequence of *P. aeruginosa*				
RKKRRQRRRGGGGITDFGIFIGL (CPP1 + AF, R23L^P^), 23 a.a.	TDFGIF IGL (15–23 a.a.)	No	TDFGIFIGL (15–23 a.a.)	No
ITDFGIFIGLGGGGSRQIKIWFQNRRMKWKK (AF + CPP2, I31K^P^), 31 a.a.	TDFGIFIG (2–9 a.a.), IKIWFQ (18–23 a.a.)	No	TDFGIFIGL (2–10 a.a.), IKIWF (18–22 a.a.)	No
RKKRRQRRRGGGGITDFGIFIGLGGGGSRQIKIWFQNRRMKWKK (CPP1 + AF + CPP2, R44K^P^), 44 a.a.	TDFGIFIG (15–22 a.a.), IKIWFQ (31–36 a.a.)	No	TDFGIFIGL (15–23 a.a.), IKIWF (31–35 a.a.)	No

^§^ Bold type denotes fragments of the peptide sequence that are the intersection of at least two programs.

**Table 2 ijms-24-16723-t002:** Results of testing peptides against various strains of pathogenic microorganisms.

Tested Peptide	Microorganisms	Peptide Activity in Liquid Medium after 15 h of Incubation (Gentamicin as Positive Control), µM, Bacteria Strain	Peptide Activity in Liquid Medium after 24 h of Incubation (Meropenem as a Positive Control), µM, Bacteria Strain
R23I^T^ based on the sequence S1 from *T. thermophilus*	*S. aureus*	12 µM, 209P strain	>3740 µM, 129B strain
	MRSA	>12 µM, ATCC 43300 strain	>3740 µM, SA 180-F strain
	*P. aeruginosa*	>12 µM, ATCC 28753	>3740 µM, 2943 strain
	*E. coli*	>12 µM, K12 strain	>3740 µM, MG1655 strain
V31K^T^ based on the sequence S1 from *T. thermophilus*	*S. aureus*	>12 µM, 209P strain	>2730 µM, 129B strain
	MRSA	>12 µM, ATCC 43300 strain	>2730 µM, SA 180-F strain
	*P. aeruginosa*	>12 µM, ATCC 28753	>2730 µM, 2943 strain
	*E. coli*	>12 µM, K12 strain	>2730 µM, MG1655 strain
I10L^P^ (ITDFGIFIGL) based on the sequence S1 from *P. aeruginosa*	*S. aureus*	913 µM, 209P strain	No data
	MRSA	913 µM, ATCC 43300 strain	No data
	*P. aeruginosa*	No data	No data
	*E. coli*	913 µM, K12 strain	No data
R23L^P^ based on the sequence S1 from *P. aeruginosa*	*S. aureus*	12 µM, 209P strain	>3780 µM, 129B strain
	MRSA	12 µM, ATCC 43300 strain	>3780 µM, SA 180-F strain
	*P. aeruginosa*	12 µM, ATCC 28753	≥3780 µM, 2943 strain
	*E. coli*	12 µM, K12 strain	≥3780 µM, MG1655 strain
I31K^P^ based on the sequence S1 from *P. aeruginosa*	*S. aureus*	>12 µM, 209P strain	>2750 µM, 129B strain
	MRSA	>12 µM, ATCC 43300 strain	>2750 µM, SA 180-F strain
	*P. aeruginosa*	>12 µM, ATCC 28753	>2750 µM, 2943 strain
	*E. coli*	>12 µM, K12 strain	>2750 µM, MG1655 strain
R44K^P^ based on the sequence S1 from *P. aeruginosa*	*S. aureus*	12 µM, 209P strain	>1930 µM, 129B strain
	MRSA	12 µM, ATCC 43300 strain	>1930 µM, SA 180-F strain
	*P. aeruginosa*	6 µM, ATCC 28753	≥1930 µM, 2943 strain
	*E. coli*	>12 µM, K12 strain	>1930 µM, MG1655 strain

## Data Availability

Data is contained within the article and Appendix A.

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
