# Peer review of "Enhancing the Antimicrobial Properties of Peptides through Cell-Penetrating Peptide Conjugation: A Comprehensive Assessment"

_ijms, 2023, doi:10.3390/ijms242316723_

Round 1
Reviewer 1 Report
Comments and Suggestions for Authors
The manuscript titled “Enhancing Antimicrobial Properties of Peptides through Cell-Penetrating Peptide Conjugation: A Comprehensive Assessment” submitted to the International Journal of Molecular Sciences presents novel findings describing the antibacterial properties and non-cytotoxicity of antimicrobial peptides and cell-penetrating peptides. Both classes of short molecules have been recognised due to their therapeutic effects and their usefulness as drug candidates mainly in the context of antibiotic resistance crisis. The introduction is quite long and could be summarised focusing on the main points relevant to this study. The results are novel and promising. Figures can be improved and standardised to facilitate understanding. To sum up, the manuscript is novel and deserves to be published, but first, the authors need carefully to address some points.
1. The abstract can be improved. Quantitative data can be included.
2. The introduction can be summarised. I believe that the mechanism of action can be explored in other sections of the document.
3. Lines 146-148, 149-151. Please avoid short paragraphs, mainly when the main idea is connected to a previous or following paragraph.
4. Please standardise the colours in the figures. The same peptide is represented in different colours in the figures.
5. Why did the authors not determine the MIC and MBC values? They are more useful in this context- A table reporting these values or a heatmap is more adequate than these dose-response figures. The dose-response can be presented as supplementary material.
6. The term cells in the figures 3 and 4 represent bacteria. Cells is a general term that can leads to misunderstanding.
7. I do not understand why the results are presented separately. Gram-positive and Gram-negative can be combined in a Table or heatmap. This facilitates comparison and understanding. I do not find any justification for this choice.
8. Line 273. Why was this bilayer composition chosen? Is similar to bacteria cell membrane composition? Were the peptides more active against bacteria with this phospholipid membrane composition? More details and justification need to be provided.
9. Figure 7. A positive control needs to be included. Triton-x or melittin?
10. Figure 8. A positive control is missing.
11. Lines 307-316. Avoid short paragraphs. If the same idea is being presented, please combine them. If not, expand the main idea and provide more details.
12. Authors used nucleated cells to access the toxicity profiles of peptides. In general, based on the data presented, the peptides are not toxic to human cells. However, to have a clearer idea of systemic toxicity I recommend using the traditional haemolytic assay employing red blood cells. If there is not enough material to perform this experiment, authors can use free-available in silico tools. This review “Traditional and Computational Screening of Non-Toxic Peptides and Approaches to Improving Selectivity” details several easy-to-use computational approaches to evaluate this cytotoxicity. I recommend the authors to explore them and discuss the results in the light of this review article. The findings can also allow to access the performance of in silico tools, which can assist in the discovery of novel selective antibacterial molecules.
13. How many bacteria were employed in the peptide susceptibility assays? How was this number determined? How many times was this experiment performed?
14. Lines 480. How many cells were employed in this viability assay?
15. Conclusions need to be summarised and focused on the main findings of the study and its perpectives.
Reviewer 2 Report
Comments and Suggestions for Authors
Please see attached file

Minor editing of the English language recommended.
Reviewer 3 Report
Comments and Suggestions for Authors
This manuscript demonstrated a nice study by enhancing the antimicrobial properties of peptides through cell penetrating peptide conjugation: a comprehensive assessment. there are a few comments to be addressed.
1. in the introduction of chemical modification of peptide antibiotics, recent summaries should be included, such as Ann. N.Y. Acad. Sci., 2020, 1459: 86-105. https://doi.org/10.1111/nyas.14282; Chem. Soc. Rev., 2021, 50, 4932-4973 https://doi.org/10.1039/D0CS01026J. Peptides, Volume 146, December 2021, 170666, https://doi.org/10.1016/j.peptides.2021.170666
2. for the control of antibacterial assay, why choose the gentamicin concentration of 200uM?
3. For the lipid bilayers, why use PE/PG as 50:50 ratio?
Comments on the Quality of English Languagesee comments
Round 2
Reviewer 1 Report
Comments and Suggestions for Authors
The authors have carefully revised the manuscript. I have no major concerns about the technical aspects of this nice research, but rather I have a few recommendations for improving the clarity of the manuscript and enriching the discussion section, highlighting a relevant aspect addressed by the authors.
1. The prediction of peptide-induced hemolysis using in silico tools must be included in section 4.7. I suggested expanding this section including in silico and in vitro approaches. The authors have already included details of in vitro assays. I also recommend the use of more than one tool. Sometimes, there are differences in the prediction.
2. Authors must explore the potential erythrocyte lysis effect caused by the peptides in the discussion section. Intriguingly, this is a very interesting result, because it contradicts the in vitro assay. Of course, the authors used different cell lines, but this result suggests the importance of using a different panel of cells and also the potential use of AI-driven approaches. On the other hand, many tools are biased and they are dependent on the database as discussed in the review paper “Traditional and Computational Screening of Non-Toxic Peptides and Approaches to Improving Selectivity”. I recommend using this paper to discuss the discrepancy between assays with fibroblast and red blood cells and also the possibility of wrong predictions of some hemolytic tools. Please discuss these relevant aspects in the discussion section.
3. The authors have added a new sentence: “The culture of normal human fibroblasts was obtained from the skin areas of the inner surface of the forearm of healthy volunteers after their consent using the explant method.” Please give details of Ethical approval.
4. The selectivity of the peptides and their systemic toxicity, particularly against red blood cells, which are essential in the body is a pending task. In this sense, authors must also include this as a perspective in the conclusion section, mainly based on the in silico prediction.
Reviewer 2 Report
Comments and Suggestions for Authors
Many minor changes have been performed in a satisfying way.
Some major uncertainties, however, remain and have not yet been solved.
They are listed in the following:
Comparison between the combined peptides and the plain AMPs (AF) with the help of data from previous studies. A table in supplement no. 4 has been added, which is good. These data from previous studies, however, are provided with a different unit (mg/ml). Data in the present publication are provided in µM. Therefore, they are not comparable for the reader.
A table must be provided where the same unit is used to make all the datasets of plain AF and its combination with CPP1 and CPP2 comparable and judgeable. For some strains, the data are present so that they can be added easily in the table.
The resulting conclusions of the comparison should be discussed in the discussion section.
There is no definition provided for ‘antibacterial efficacy’. Please specify.
There is still no detailed statistical evaluation provided for single concentrations in figures 3 and 4, and also not in supplemental figure 2.
ANOVA should be done for single concentrations of peptides in these figures.
Again, provide significances compared to the control with * (p < 0.05) and ** (p<0.01) as usual for single concentrations / columns in case of statistically significant differences. Remove the term ‘near-zero values’ at best - or provide a clear definition for this term and do not use an asterisk but another sign or superscript letter for it.
Provide results of ANOVA testing in supplemental figure 1 at least for time points 10 and 15 hours.
For reliable statistics, it would be preferable to perform at least three independent repetitions instead of two in this study. This may not be obligatory, although desirable for the present study, but in any case a useful hint for future studies.
Testing higher concentrations against ‘susceptible’ strains presented in table 2 and possibly against human cells (figure 6) should be discussed. The answers given to the issues of the reviewer in this regard are imprecise and circumventive.
If clinically achievable concentrations are unknown so far, this should be mentioned. If there are hints from studies, please discuss. It also should be mentioned that it is a limitation of the study that threshold concentrations of activity against several bacteria used remain unknown and should be tested in the future etc.
Chapter 4.4. of the methods is still unclear regarding the resistance of strains. Please clearly describe against what each strain was susceptible (AMPs ? antibiotics ?) or resistant. MRSA has a clear definition. The MRSA descriptions are still more confusing than instructive. Were these really MRSA strains ?
Tests of strains with different media in different institutes.
It should be mentioned as a limitation of the study that this was not uniform and the influence of the media remains to be checked. If there are any results on this question available, please mention in short form and discuss on your present state of research.
Please remove the term ‘cells’ in supplemental figures 1 and 2 since it deals with bacteria.
